# The Development of a Novel Thin Film Test Method to Evaluate the Rain Erosion Resistance of Polyaspartate-Based Leading Edge Protection Coatings

Stephen M. Jones [1],* , Nadine Rehfeld [2], Claus Schreiner [2] and Kirsten Dyer [1]

1   Offshore Renewable Energy Catapult, Offshore House, Albert Street, Blyth NE24 1LZ, UK; kirsten.dyer@ore.catapult.org.uk
2   Department Paint Technology, Fraunhofer Institute for Manufacturing Technology and Advanced Materials IFAM, 28359 Bremen, Germany; nadine.rehfeld@ifam.fraunhofer.de (N.R.); claus.schreiner@ifam.fraunhofer.de (C.S.)
*   Correspondence: stephen.jones@ore.catapult.org.uk

**Abstract:** The relationship between the bulk thermomechanical properties and rain erosion resistance of development polyaspartate-based coatings as candidate leading edge protection (LEP) materials for wind turbine blades is investigated by the combined application of dynamic mechanical analysis (DMA) and rain erosion testing (RET) within a novel test method (DMA+RET). This method introduces the use of DMA+RET to both monitor the change in thermomechanical properties with respect to raindrop impact and subsequently rationalise differences in rain erosion resistance between coating formulations of comparable composition. The application of this combined process has demonstrated the importance of relatively high viscoelastic moduli at increased strain rates and creep recovery after RET as key material properties to be considered for LEP material development, whereas previous research presented in the scientific literature has primarily focussed on the use of routine characterisation procedures by tensile testing or stand-alone DMA to evaluate coating formulations prior to rain erosion testing. This journal article therefore presents a novel method to evaluate key material properties relevant to rain erosion resistance before and after subjection to raindrop impact using standard ASTM G73 RET equipment. The test method is demonstrated on a novel polyaspartate-based coating, PA-U, that exhibits notable rain erosion resistance in comparison to commercial LEP products. PA-U exhibited negligible mass loss after 30 h of rain erosion testing and favourable thermomechanical properties ($E'' = 35$ MPa at critical strain; equilibrium recoverable compliance of 0.05 MPa$^{-1}$) in comparison to alternative formulations.

**Keywords:** creep recovery; dynamic mechanical analysis; leading edge erosion; leading edge protection; polyaspartate

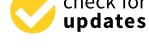



## 1. Introduction

The action of raindrop impact on turbine blades to initiate leading edge erosion (LEE) is well documented [1–3] and represents a significant issue within the wind turbine industry. LEE is reported [4] to reduce the aerodynamic profile of the blade that subsequently lowers the Annual Energy Production (AEP) of a turbine in operation. The maintenance and repair costs associated with restoring optimum blade performance to maximise the Levelised Cost of Energy (LCoE) attained by the wind turbine have been estimated at up to $200,000 per blade [5] and is required to occur every 2–5 years [6].

A key contributing factor to the magnitude of LEE on wind turbine blades is the selection and lifetime performance of the leading-edge protection (LEP) material that is installed on the blade to mitigate losses in AEP and LCoE. Both development and commercial LEP materials typically comprise of thermoplastic polyurethane [7–9] and/or polyurea [10] coatings due to their relatively high rain erosion resistance, adhesive properties and inexpensive

cost when compared to alternative polymeric materials. Similar formulations are available in tape and preformed shield format [1]. The LEP coating may also incorporate additives including pigments and/or fillers at <15% in terms of the total mass of the composition. A pigment or colouring agent is added to impart the desired colour to the LEP material, whereas a filler may improve coating properties, e.g., weathering performance or durability, in addition to reducing the content and cost of the polymeric binder material [11,12].

The current recommended practice for testing of rotor blade erosion protection systems is defined by DNVGL-RP-0171 [13]. However, within this guideline, there is no testing dedicated to relating material properties and behaviour to the subsequent observations after erosion has occurred. As such, material property–erosion correlations to date have largely been documented by research publications to compare the relative measure of a material property tested independently against the erosion resistance of that material. It therefore remains imperative to find new ways to effectively screen candidate LEP formulations with material property changes and impact from rain droplets concurrently to increase the understanding of formulation effects on rain erosion resistance. Faster, cheaper test techniques will also enable solution development for the wind industry.

As outlined in Table 1, a number of studies [14–18] have inferred correlations between the surface or bulk material properties of a LEP coating with rain erosion resistance. Such processes have traditionally relied upon a destructive (tensile test/nano-indentation) or non-destructive (dynamic mechanical analysis (DMA)) characterisation method being performed on a flat material sample. Separate, often curved, samples of the material are then subjected to raindrop impacts via a standard accelerated rain erosion test (RET). The LEP samples may then be ranked in order of their rain erosion resistance and compared against identifiable trends observed in the characterisation data. This process is useful for identifying key material properties that may be related to erosion resistance but does not allow for investigations or explanations of changes in material behaviour during RET connected to formulation differences.

The correlations listed in Table 1 suggest that the rain erosion resistance of LEP materials may be considered somewhat analogous to the material behaviour of elastomers, in that the product should exist in application above the glass transition temperature, $T_g$, to permit molecular conformation and also exhibit relatively high Poisson ratio and elongation to break (ETB) values. It is also proposed in some cases that a LEP material should exhibit relatively low values for Young's modulus and hardness. However, it is notable that the listed material property–erosion correlations are not universal between several studies that involve different material classes, e.g., coatings, thermoplastics and also metallics, and that several of the correlations are based upon tensile testing at low strain rates of ca. $0.001–0.01 \text{ s}^{-1}$ compared to that of erosion. In particular, the research presented by Katsivalis et al. [18] demonstrates the difficulty so far in proposing material property–erosion correlations between LEP materials consisting of similar polymer functionalities (TPU), due to the number of potential material properties for consideration.

The importance of material recovery has also been highlighted as an essential property for LEP materials. As previously outlined by O'Carroll et al. [19], it is preferable for a material to dampen raindrop impact energy and recover to an original state to avoid permanent plastic deformation [20,21]. This hypothesis appears to be particularly valid with respect to portraying a more accurate representation of the method of raindrop impact [22] and simulation of rainfall intensity patterns upon a blade surface. O'Carroll used nanoindentation with a Berkovitch indenter as an analytical technique to study the short-term (elastic) and long-term (viscoelastic) recovery of several thermoplastic polymers. An increase in short-term recovery, on the timescale of approximately 1 minute, was observed to correlate with an improvement in rain erosion resistance for the particular materials tested. However, it is anticipated that impact and recovery time scales in terms of erosion are in the order of microseconds, i.e., based on the frequency of raindrop impacts, and absolute property values change with the nature of the impacter and event so rain droplets are required to obtain a true correlation.

**Table 1.** Selected summary of previous inferred correlations between material properties and rain erosion resistance [14–18].

| Reference | Rain Erosion Medium | Material Classification (Rain Erosion Resistance) | Inferred Material Property Requirements | Material Characterisation Technique |
|---|---|---|---|---|
| N. Hoksbergen et al. [14] | RET Springer model, i.e., mathematical | TPU > PAI > PEEK > PC > PBT > ABS > PTFE > PE | Increased Poisson ratio (elastomeric) | Fatigue test (Wöhler S-N) |
| A. O'Carroll [15] | WARER | PP > PE > PC > PET > PMMA | Reduced UTS | Tensile test (ASTM D638-14) |
| | | | Reduced hardness | Nanoindentation |
| | | | Increased short-term recovery | |
| | | | Reduced acoustic impedance | Ultrasonic evaluation |
| | | | Increased damping ratio | |
| G.F. Schmitt [16] | Rotating arm apparatus | TPU > PE > PA > PI | Increased ETB | Tensile test (ASTM D412) |
| H.M. Slot et al. [17] | Stationary nozzle spray | PBT > PA > PET > PP > PE > PC > PMMA | Reduced E (Young's modulus) | Tensile test |
| | | | Reduced polymer crystallinity | |
| I. Katsivalis et al. [18] | WARER | 9 × TPU | E (Young's modulus)–inconclusive | Tensile test |
| | | | Stiffness and hardness–inconclusive | Nanoindentation |
| | | | Reduced E′ (when measured at $\geq 10^6$ Hz) | DMA (frequency sweep/TTS) |

RET = rain erosion test; WARER = whirling arm rain erosion rig; TPU = thermoplastic polyurethane; PAI = polyamide-imide; PEEK = polyether ether ketone; PC = polycarbonate; PBT = polybutylene(terephthalate); ABS = acrylonitrile butadiene styrene; PTFE = polytetrafluoroethylene; PE = polyethylene; PP = polypropylene; PET = polyethylene (terephthalate); PMMA = poly(methyl methacrylate); PA = polyamide; PI = polyimide; ETB = elongation to break; DMA = dynamic mechanical analysis; TTS = time-temperature superposition.

It is therefore surmised that there remain clear discrepancies in rankings of candidate polymer materials by their relative erosion resistance according to material properties, of which some inconsistencies may be attributed to variations in erosion apparatus and specimen geometry, the conditions of erosion testing, and variations in polymer morphology, physical properties and formats. The derived correlations as listed in Table 1 between material properties and erosion resistance may therefore be valid but limited in their capacity to estimate application erosion performance between LEP coatings of comparable polymer composition or filler content under different erosion conditions and testing conditions. In addition, rain erosion testing (RET) methods are often time-intensive with experiments occurring for between approximately 2 to 200 h, with regular inspection periods and visual assessments required to monitor the progression of material erosion.

One benefit of using DMA as an analytical technique for measuring relevant thermomechanical material properties is that the testing variables of frequency and time may be incorporated into characterisation procedures. For example, Katsivalis et al. [18] and Ouachan et al. [23] utilised time-temperature superposition (TTS) methodology to obtain thermomechanical properties including storage modulus, E′, loss modulus, E″, and tan δ at frequencies up to $10^6$ Hz, which is reported [1] to be comparable against the strain rates generated during droplet impact. E′ and E″ represent the elastic and viscous components of energy that is either stored or dissipated from the sample upon the application of stress

within DMA, respectively. Tan $\delta$ represents the ratio of viscous ($E''$) to elastic ($E'$) response for a viscoelastic material which therefore gives an indication towards damping potential. Previous research (Table 1) [18] has suggested that a relatively low $E'$ and improved damping ability (tan $\delta$) is preferable for maximising rain erosion resistance.

It is also reasonable to consider DMA as a non-destructive characterisation method, assuming the measurements are performed within the linear viscoelastic region (LVER) thus avoid disrupting the microstructure of the material. Finally, creep recovery experiments may be performed using DMA to provide an indication of material elasticity and short-term recovery, which will be presented as a novel experimental procedure in context to evaluating rain erosion resistance within Section 2.2.2 and Supplementary Materials.

This journal article aims to provide LEP material property–erosion correlations by means of a novel combined DMA+RET test method. In practical terms, a thin film of LEP material with dimensions of ca. 50 mm $\times$ 10 mm $\times$ 0.1–1 mm is characterised by DMA before being attached to a bespoke test specimen for RET exposure in an industry standard test rig, and is then detached after a defined period of RET. The thin film of LEP material is then subjected to several DMA experimental procedures to obtain key thermomechanical properties before being reattached to the test specimen and further RET as desired. This iterative cycle therefore combines intermittent material characterisation by DMA in combination with accelerated RET, in contrast to previous research which has solely obtained material property data prior to RET. This test method therefore enables the monitoring of several key material properties with respect to raindrop impact, which provides greater information regarding property deterioration and the potential mechanism of rain erosion in the LEP under test. The evaluation of rain erosion resistance is achieved at reduced time and cost (the procedure takes less than 1 day whatever the erosion resistance of the material) to obtain critical information on key material properties of the LEP which correlate to behaviour observations in full scale RET and LEP formulation. The application and validation of the novel combined DMA+RET test method (performed by Offshore Renewable Energy Catapult) is herein applied to the evaluation of rain erosion resistance of several development polyaspartate-based coatings supplied by Fraunhofer IFAM.

## 2. Materials and Methods

### 2.1. Sample Definitions

Three polyaspartate-based coating formulations of equivalent polymer composition were prepared for this investigation as outlined in Table 2. Further information regarding the initial screening process and tensile analysis [24] of the polyaspartate binder is detailed in Supplementary Materials. Each coating was manufactured in two formats corresponding to the subsequent testing regime where RET is defined as full scale rain erosion testing using the rain erosion tester in coated composite specimen format to DNVGL-RP-0171, and the novel combined DMA+RET test method is defined as the combined use of DMA and RET testing in the rain erosion tester in free film format that are attachable/detachable from a bespoke composite leading edge specimen. A cross-section scheme for each experimental setup is illustrated in Figure 1. The polyaspartate-based coating coded PA-U contained no inorganic filler, whereas the coatings coded PA-FB and PA-FS contain 6–7 wt% of an undisclosed inorganic filler, but are deposited via brush and spray application, respectively. All coating materials were prepared using standard mixing and dispersion techniques (Dissolver DISPERMAT CV3-PLUS (VMA-Getzmann, Germany) with shear blade).

### 2.2. Experimental Procedures

#### 2.2.1. RET

The sample preparation method for evaluation of rain erosion resistance using the rain erosion tester for coated composite leading edge specimens is illustrated in Figures 2 and 3. Sikaforce 7800 Red (Sika AG, Baar, Switzerland) was first applied as a suitable wind turbine blade filler material at a thickness of approx. 1 mm on the surface of curved glass/epoxy composite specimens of approximate dimensions of 80 mm $\times$ 450 mm (Olsen Wings A/S,

Odder, Denmark), and cured according to the stated conditions on the product datasheet (30 minutes at 23 °C). A polyaspartate-based coating was then applied by one of several methods (pour, brushing or spray) and cured for 16 h at 50 °C.

**Table 2.** Sample information for polyaspartate-based coating formulations.

| Analysis Method | RET | | | Novel Thin Film Test Method (Combined DMA+RET) | | |
|---|---|---|---|---|---|---|
| Sample code | PA-U | PA-FB | PA-FS | PA-U | PA-FB | PA-FS |
| Manufacturing method | Poured | Brush | Spray | Cast | Brush | Spray |
| Inorganic filler content (wt%) | - | 6–7 | 6–7 | - | 6–7 | 6–7 |
| Colour | Transparent | Grey | Grey | Transparent | Grey | Grey |
| Thickness (mm) | Approx. 0.5 | 0.5–0.7 | Approx. 0.5 | 0.24 | 0.57 | 0.32 |
| Dimensions (w × l) | Approx. 80 mm × 450 mm | | | Approx. 50 mm × 10 mm | | |
| Format | Coated composite specimen | | | Free film | | |

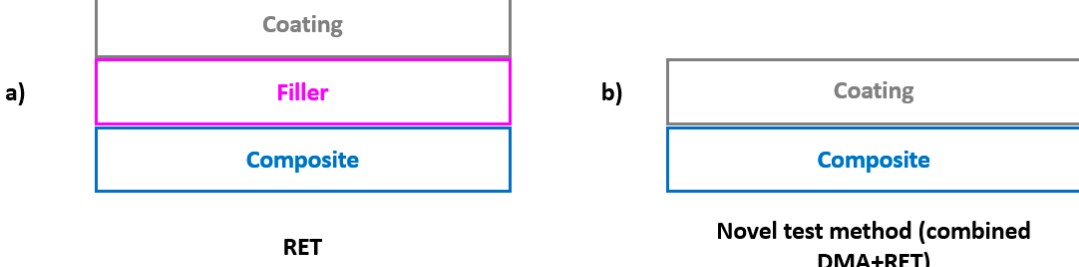

**Figure 1.** Cross-section of each experimental setup used in this investigation where (**a**) RET = a coated composite leading edge test specimen; and (**b**) the combined DMA+RET test method = a detachable free film on a composite leading edge test specimen.

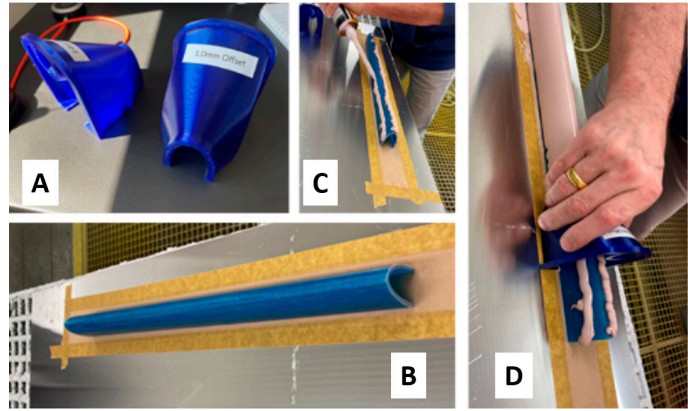

**Figure 2.** Preparation of coated rain erosion test specimens where (**A**) = 3D printed smoothing tool; (**B**) = composite specimen; (**C**) = application of filler (Sikaforce 7800 Red); (**D**) = filler smoothing.

The rain erosion resistance of polyaspartate-based coatings was evaluated using the standard wind industry rain erosion tester (R&D A/S, Hinnerup, Denmark) at Offshore Renewable Energy Catapult in accordance with the DNVGL-RP-0171 guideline [13] for testing of rotor blade erosion protection systems. This equipment (Figure 4) comprises a three-bladed rotor with test specimen holders that are subjected to repeated rain droplet impacts to simulate rainfall intensity. All experiments were performed at a rotational velocity of 1000 rpm (which provides local impact velocities of approximately 84, 105 and 125 m/s at the root, centre and tip of the specimen, respectively) with a mean droplet size

diameter of 2.50 $\pm$ 0.04 mm and flow rate of 55 L hour$^{-1}$. Unless otherwise stated, the specimens were visually inspected every hour by removing the samples and recording the erosion defects against linear velocity, until exposure of the underlying composite structure was obtained due to significant coating erosion and mass removal.

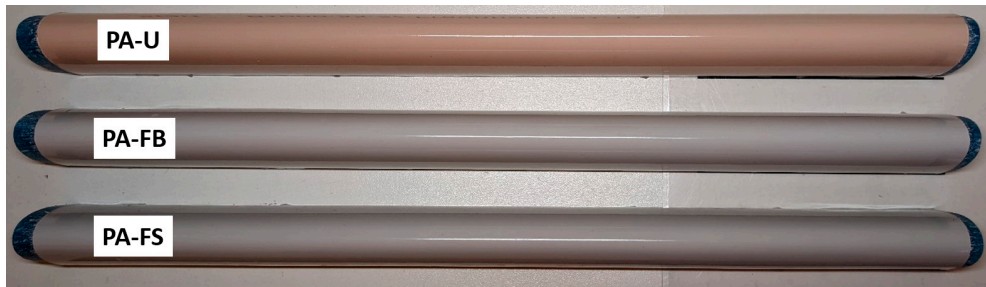

**Figure 3.** Representative rain erosion test specimens for the rain erosion tester.

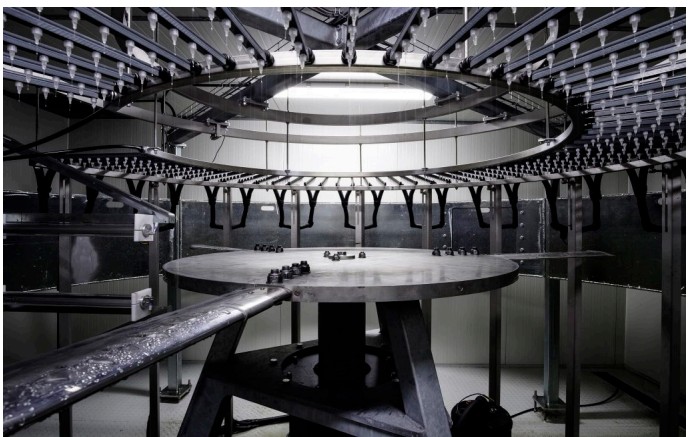

**Figure 4.** The rain erosion tester (R&D A/S, Hinnerup, Denmark) at Offshore Renewable Energy Catapult, U.K (Source: Offshore Renewable Energy Catapult).

2.2.2. Combined DMA and RET

All polyaspartate-based coatings were also cast in free film format upon a 2 mm thick polyethylene film and cured for 16 h at 50 °C. Samples with approximate dimensions of 50 mm length × 10 mm width were then obtained using a Freehand Strip Cutter (RDM Test Equipment, Stortford, UK.). It is noted that the free film sample dimensions are commensurate with the recommended practice provided by the equipment manufacturer and therefore defined as standard within the context of performing DMA testing.

DMA was performed using a DMA 850 (TA Instruments, New Castle, DE, USA) with the modular tension clamp attachment. All DMA experimental procedures featured an initial conditioning step, and the temperature was equilibrated at the defined value for 5 minutes. Table 3 details the test parameters used for DMA oscillatory experimental procedures. The modular tension clamp was selected for DMA procedures within this research for two reasons. Firstly, it is recommended practice by the equipment manufacturer to utilise this fixture when measuring the thermomechanical properties of films in the thickness range studied (0.25–0.6 mm), which is also comparable to the typical thicknesses of LEP coatings when applied to wind turbine blades. In addition, as reported by Zahavi et al. [25] with regard to the mechanism of droplet impact, the peak stresses incurred by a LEP coating are tensile stresses that result from the reflection of the initial compression wave. Hence, it is valid to utilise a tensile fixture for the assessment of thermomechanical properties for candidate LEP products.

**Table 3.** Test parameters for DMA oscillatory experimental procedures.

| Experimental Procedure | Oscillation Strain Sweep | Oscillation Frequency Sweep/TTS | Oscillation Temperature Ramp |
|---|---|---|---|
| Temperature (°C) | 20 | −60 to 60 at 10 °C intervals | −70 to 100 at 3 °C min$^{-1}$ |
| Amplitude (μm) | 0.1–10,000 | 20 | 20 |
| Frequency (Hz) | 1 | 0.1–10 | 1 |

Oscillation frequency sweep data acquired at 10 °C intervals between −60 and 60 °C was used to extrapolate the values of E′, E″ and tan δ across a frequency range of $10^{-2}$ to $10^{10}$ Hz by application of TTS within the TRIOS software package (v5.1.1) provided by TA Instruments. The validity of the produced TTS master curves was confirmed by shift factor analysis using the Williams-Landel-Ferry and Arrhenius models as previously reported [26], to obtain $R^2$ > 0.98 in all cases and therefore validation of the application of TTS to the materials studied. In addition, the Arrhenius activation energies for all coatings were determined in close proximity between 201–202 kJ/mol, which is consistent with previous values obtained for polymers used as LEP products [27].

DMA was also utilised to perform a creep recovery experimental procedure, which is further detailed in Supplementary Materials [28,29]. In the first step of this method, a constant force of 1.0 N was applied to each sample for a creep time of 2 minutes at 20 °C. It is noted that the applied force of 1.0 N translated to an applied stress that was within the linear viscoelastic region, LVER, for all polyaspartate-based coatings. The applied stress is then removed, which initiates instantaneous elastic recovery in the sample. Time-dependent viscoelastic recovery then proceeds at a slower rate until the end of the experimental procedure, which was defined by a recovery time of 10 minutes. All presented values for the equilibrium recoverable compliance, $J_{er}$, were obtained at the end of the recovery step after 10 min, and therefore provide an estimation of the short-term recovery and elasticity for the material sample. A recovery time period of 10 minutes was selected in order for all reversible strain to be recovered and therefore allow a steady-state value for $J_{er}$ to be achieved.

Creep recovery data acquired at 10 °C intervals between −40 and 60 °C was used to extrapolate the values of $J_{er}$ across a frequency range of $10^{-2}$ to $10^{10}$ Hz by application of TTS via the same method as previously detailed.

The analyses described above were performed on the samples before attaching the same free film of the polyaspartate coating to the bespoke composite RET specimen. The samples were exposed to the same test parameters as previously for periods of 30 minutes. Repeat DMA analyses were performed at the intermittent stop periods.

## 3. Results and Discussion

The rain erosion resistance for each polyaspartate-based coating according to the DNVGL-RP-0573 guideline [30] was observed to conform with the previously reported [31] stages of erosion progression. As illustrated in Figure 5, no visual surface damage is observable and there is no material mass loss during the initial testing stage. The end of the incubation stage is identified via the onset of observable surface defects in the form of pitting. In the second stage, the coating begins to exhibit significant and accelerating mass loss to induce surface roughness, until failure, defined as exposure of the composite laminate, occurred.

Figures 6 and 7 detail significant differences in the rate of erosion progression and mass loss between the unfilled (PA-U) and filled (PA-FB and PA-FS) polyaspartate-based coatings. The times to incubation and composite breakthrough were recorded after 2 ± 0 and 6 ± 1.5 h, respectively, for both filled coatings in comparison to after 24 ± 13 and 29 ± 3.5 h, respectively, for PA-U. Furthermore, the initiation of mass loss occurred at approximately 6 h for both filled coatings. In contrast, negligible mass loss was observed for PA-U after 30 h across all repeat experiments, which confirms that PA-U exhibits

comparable rain erosion resistance in comparison to several commercial LEP coatings tested under equivalent conditions. The particularly high standard deviation observed for the time to incubation for PA-U is due to the generation of an incubation point after 9 h of RET for one test sample (3 repeat tests were performed), whereas the other two test samples observed no incubation points after 31 h of RET. This phenomenon suggests that the erosion progression mechanism for PA-U has proceeded via a single defect-driven mechanism, in contrast to the multiple pits that have formed as a function of testing time for PA-FB and PA-FS.

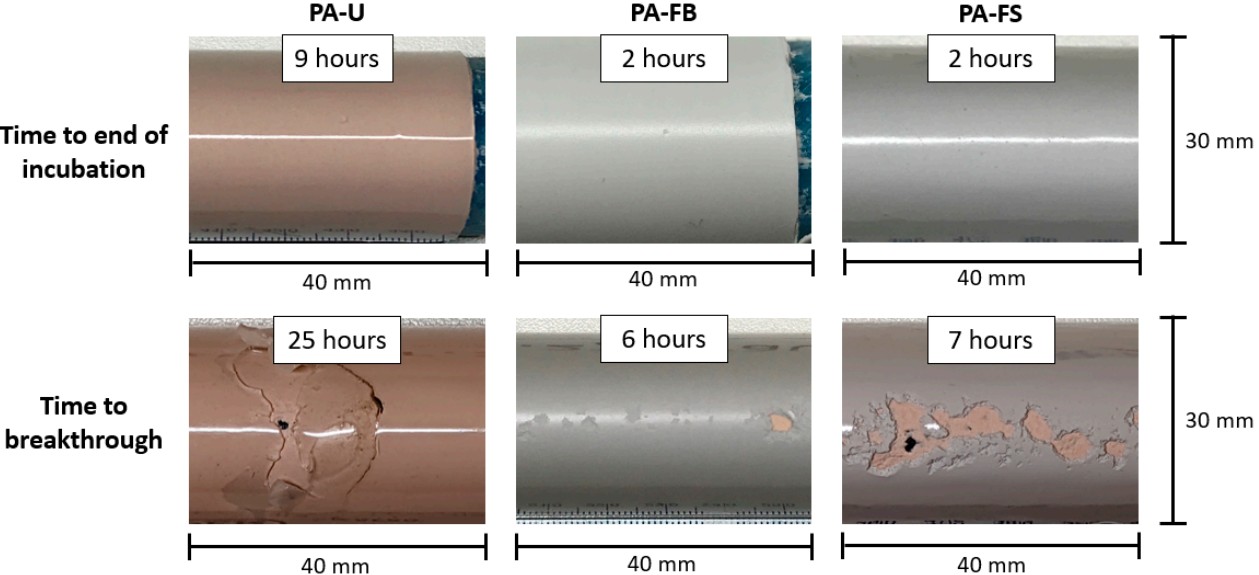

**Figure 5.** Rain erosion progression for selected tests of all polyaspartate-based coatings when subjected to raindrop impact via the rain erosion tester.

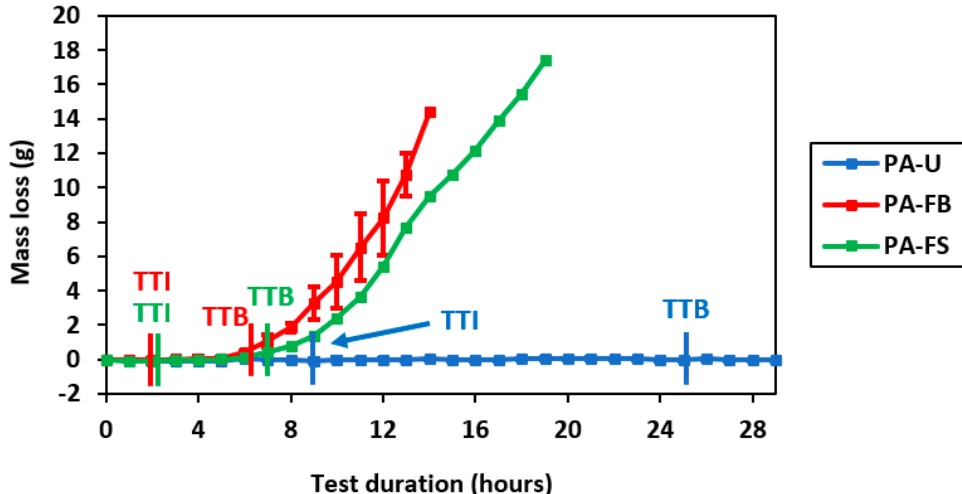

**Figure 6.** Average cumulative mass loss curve for all polyaspartate-based coatings where TTI = time to incubation; and TTB = time to breakthrough, for each coating.

　　In order to infer correlations between the material properties and rain erosion resistance of the polyaspartate-based coatings, several thermomechanical properties including $T_g$, $E'$ and $E''$ were measured by standard DMA experimental procedures (oscillation temperature ramp and oscillation frequency sweep) for each coating (Table 4) as described in Section 1. However, there was little observable difference in $T_g$ between all coatings when considering definition either by the $E''$ or tan δ peak in terms of the absolute temperature recorded. A reduction in $E'$ and $E''$ was observed at 1 Hz upon the inclusion of filler for

PA-FB and PA-FS when obtained by a frequency sweep experimental procedure at 20 °C, yet the tan δ values for each filled coating are greater than observed for PA-U. The apparent positive correlation between an increased E′ and improved rain erosion resistance is notable as this is in direct contrast to conclusions generated within previous research [16,18]. The observations produced from both DMA experimental procedures demonstrate that said methods are insufficient to account for marked differences in rain erosion resistance between the unfilled and filled polyaspartate-based coatings.

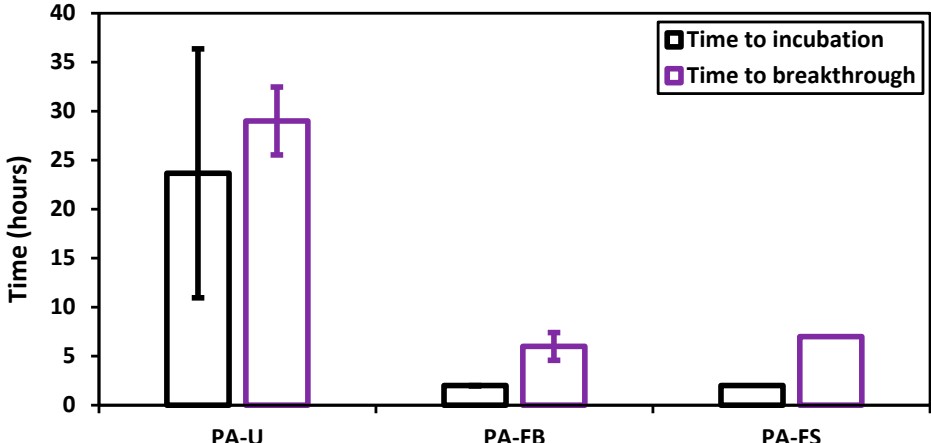

**Figure 7.** Erosion progression data for all polyaspartate-based coatings.

**Table 4.** Thermomechanical properties of polyaspartate-based coatings obtained by standard DMA experimental procedures.

| Coating | $T_g$ (E″ Peak, °C) [a] | $T_g$ (tan δ Peak, °C) [a] | E′ (MPa) [b] | E″ (MPa) [b] | Tan δ [b] |
|---------|--------|--------|--------|--------|--------|
| PA-U | −21.6 | 36.1 | 105.4 | 37.9 | 0.36 |
| PA-FB | −22.6 | 31.7 | 39.0 | 19.9 | 0.48 |
| PA-FS | −21.7 | 37.1 | 79.0 | 30.0 | 0.38 |

[a] Obtained via DMA temperature ramp at 3 °C min$^{-1}$. [b] Obtained via DMA frequency sweep at 1 Hz, 20 °C.

The effect of DMA testing frequency upon the values of E′, E″ and tan δ for the polyaspartate-based coatings was further investigated by application of the TTS principle to frequency sweep data acquired at 0.1 to 10 Hz between −60 and 60 °C. As illustrated in Figure 8, an increase in E″ is observed with respect to increasing frequency that proposes improved damping ability at time periods relevant for the method of raindrop impact (ca. $10^6$ Hz). However, given that E′ also increases to a greater extent at higher frequencies as perhaps expected for viscoelastic materials, the tan δ values for all coatings decrease over the same period. The observed increase in E′ and E″ with respect to the test frequency may be attributed to an effective increase in material stiffness and damping at higher frequencies, whereby the molecular chain motion is effectively decreased relative to the timescale of the imparted force. This phenomenon for E′ and E″ has been previously observed on numerous occasions for viscoelastic solids [32,33]. There is also negligible difference in the values of E′, E″ or tan δ for each coating that may account for the discrepancies in rain erosion resistance at frequencies of approximately $10^6$ Hz, which was previously detailed to be comparable against the strain rates generated during droplet impact. The application of TTS methodology by acquisition of frequency sweep data has therefore also proven to be inconclusive with respect to providing suitable correlations between the bulk material properties and rain erosion resistance of the polyaspartate-based coatings.

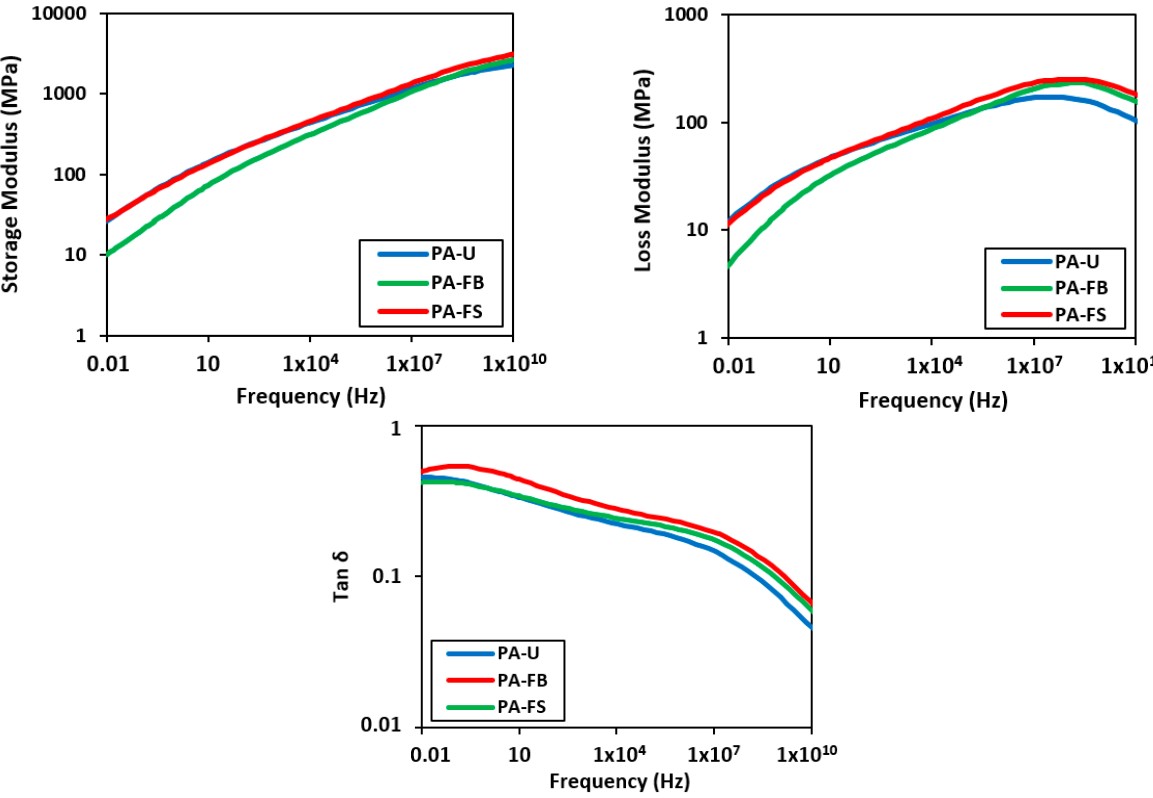

**Figure 8.** TTS master curves for E′ (**top-left**), E″ (**top-right**) and tan δ (**bottom**) as a function of frequency (Hz) at a reference temperature of 20 °C.

The utilisation of experimental procedures by the combined DMA+RET test method was then performed to monitor the progression of bulk material properties with respect to periods of repeated raindrop impact via RET. In this setup, several thermomechanical properties of each coating with dimensions ca. 50 mm × 10 mm × 0.2–0.6 mm, were obtained by DMA after curing and 30 minutes of RET, respectively.

The values of E′, E″ and tan δ for all polyaspartate-based coatings were first remeasured by DMA at 1 Hz and 20 °C using the frequency sweep experimental procedure, and again after 30 minutes of RET, for comparison against the baseline measurements listed in Table 4. It is observed in Figure 9 that the single notable change in viscoelastic properties for any coating after 30 minutes of rain erosion testing is an approximate 20% decrease in E′ and E″ for PA-U, when using this DMA experimental procedure (oscillation frequency sweep). The values for E′, E″, and tan δ for both PA-U and PA-FS remain comparable after 30 minutes of RET exposure despite the observed differences in rain erosion resistance between formulations.

The viscoelastic properties of all polyaspartate-based coatings were therefore considered at increased oscillation stresses by the combined DMA+RET test method to induce oscillation strains of up to 10% via a modified oscillation strain sweep at 1 Hz and 20 °C. This aimed to obtain thermomechanical properties both within and outside the linear viscoelastic region (LVER) of a test sample at a constant amplitude, in order to more accurately reflect the range in tensile stresses that may be imparted upon a LEP coating after raindrop impact. The end of the LVER was defined at 95% of the maximum E′ value [34]. All coatings exhibited a decrease in E′ and E″ with respect to increasing oscillation strain. This phenomenon has been categorised [35] as the Payne effect and may be attributed to the deformation and breakage of intermolecular forces between polymer and filler regions. Analogous behaviour for thermoplastic polyurethanes has also been previously reported by Schaefer et al. [36] and Strankowski [37]. The reduction in E′ with respect to increasing oscillation strain for an unfilled polyaspartate is proposed to occur from irreversible struc-

turing of the morphology and hard-segment orientation as detailed by McLean et al. [38] and Abouzahr et al. [39] for thermoplastic polyurethanes. The variation in E″ is reported to occur from the continuous transition between recoverable and unrecoverable acquisition of strain at increasing strain amplitudes [40].

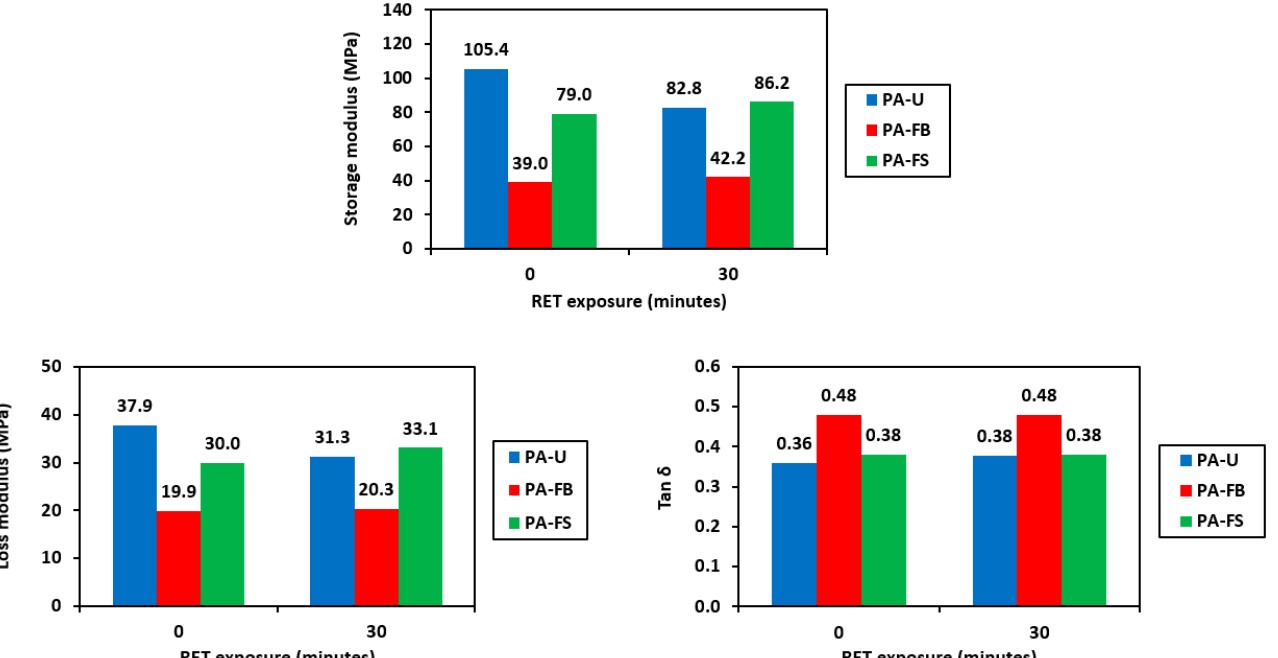

**Figure 9.** Comparison of E′ (**top**), E″ (**left**) and tan δ (**right**) at 1 Hz, 20 °C, for polyaspartate-based coatings before and after 30 min of rain erosion testing.

The addition of filler for PA-FB and PA-FS is observed to both reduce the critical strain, $\gamma_c$, denoting the end of the LVER, and the value for each viscoelastic moduli (E′ and E″) at which $\gamma_c$ occurs (Table 5). It may therefore be concluded that the inclusion of a filler has effectively shortened the LVER and intensified the strain dependency relative to PA-U, as previously reviewed by Barrera et al. [41] for the inclusion of fly ash in rubber. E′ and E″ are also observed to decrease further across for all coatings at measurable oscillation strains following exposure to RET for 30 min. As illustrated in Figure 10, the reduction in E″ was more pronounced for both filled coatings, PA-FB and PA-FS at the end of LVER in comparison to PA-U (ca. 30% vs. 10%). These observations indicate that the damping ability for both filled coatings is markedly reduced at all oscillation strain values, and that the coating microstructure containing fillers deteriorates at a quicker rate upon raindrop impact. Furthermore, the presented data suggests that this DMA experimental procedure should be performed during the evaluation of LEP materials, given that the combined use of DMA and RET has demonstrated differences in material properties, after exposure to RET, that were not previously observable when using the previously documented oscillation frequency sweep procedure.

**Table 5.** Values for E′ and critical strain, $\gamma_c$, at the end of the linear viscoelastic region (LVER) for each polyaspartate-based coating.

| Coating | E′ (MPa) | | $\gamma_c$ (%) | |
|---|---|---|---|---|
| | **0 M** | **30 M** | **0 M** | **30 M** |
| PA-U | 87.6 | 72.0 | 0.87 | 1.13 |
| PA-FB | 58.3 | 36.2 | 0.45 | 0.47 |
| PA-FS | 80.7 | 54.6 | 0.44 | 0.51 |

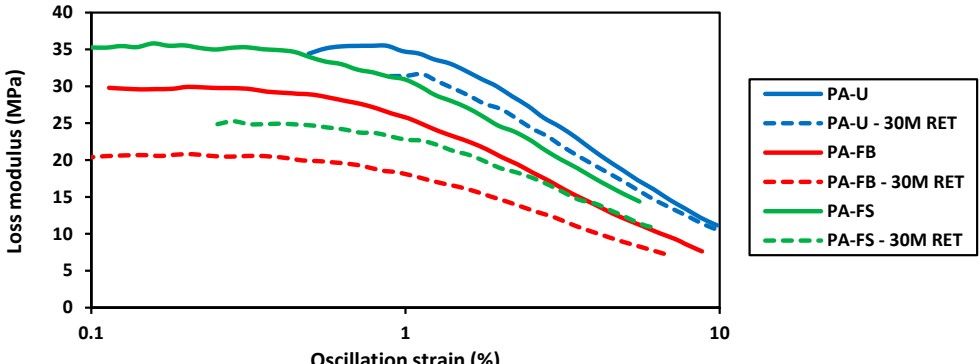

**Figure 10.** Comparison of E″ at increased oscillation strain for polyaspartate-based coatings before and after 30 min of rain erosion testing.

The greater reduction in viscoelastic moduli at increased oscillation strains for the filled polyaspartate-based coatings, PA-FB and PA-FS, also signified a time-dependent variable that should be considered for material characterisation, given that all film coating samples were subjected to the equivalent raindrop impact frequencies over a 30 minute RET period. Therefore, all coatings were subjected to creep recovery experimental procedures by DMA (as outlined in Section 2.2.2) to estimate short-term recovery and elasticity properties.

A decrease in the value of the equilibrium recoverable compliance, $J_{er}$, is inversely correlated to an estimation of material elasticity [29,42]. The graph presented in Figure 11 therefore confirms that PA-U exhibits the highest elasticity and potential for short-term recovery in contrast to both filled coatings, PA-FB and PA-FS. This trend also indicates that the lowest proportion of non-recoverable strain, or plastic deformation, was observed by PA-U. A reduction in $J_{er}$ upon the inclusion of inorganic fillers to thermoplastic polymers has also been previously reported [43–45]. It is probable that the addition of inorganic filler is located predominantly within the soft phase of the polyaspartate coating. Therefore, the relaxation time relating to the disentanglement of polymer chains occurring after an applied strain is increased, which prevents effective rearrangement and recovery prior to subsequent raindrop impacts. All coatings exhibited an increase in $J_{er}$ following 30 minutes of RET, which suggests that the method of raindrop impact has caused damage to the short-term recovery properties of each coating, and that a lower value for $J_{er}$ is preferable prior to rain droplet exposure. It is also recommended that this DMA experimental procedure is herein utilised as an experimental procedure for the evaluation of candidate LEP products, since coating elasticity and short-term recovery have been identified as material properties that will vary with respect to raindrop impact following the application of the novel thin film test method.

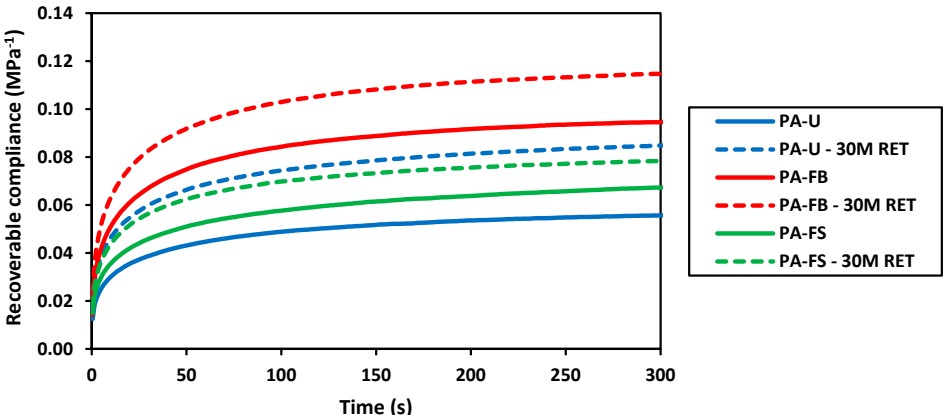

**Figure 11.** Progression of the equilibrium recoverable compliance, $J_{er}$, as determined by DMA creep recovery experiments for all polyaspartate-based coatings before and after 30 minutes of rain erosion testing.

TTS methodology was also employed to further investigate $J_{er}$ as a function of frequency (Figure 12). The lowest value for $J_{er}$ was observed for PA-U across all frequencies between $10^{-2}$ and $10^{10}$ Hz, which therefore predicts improved material recovery over time periods of $10^{-10}$ to $10^{2}$ s in comparison to PA-FS and PA-FB. It is therefore reasonable to conclude that PA-U exhibits the highest rain erosion resistance of the coating systems studied due to possessing increased viscoelastic moduli (E$'$ and E$''$) across a range of induced strains and superior short-term recovery properties across all frequencies.

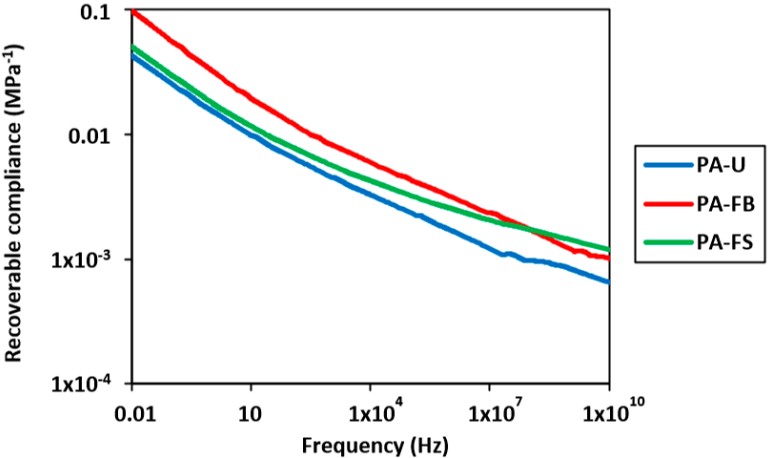

**Figure 12.** TTS master curves for the equilibrium recoverable compliance, $J_{er}$, of each polyaspartate-based coating as a function of frequency (Hz) at a reference temperature of 20 °C.

## 4. Conclusions

The primary outcome of this research is the development of experimental procedures that are able to provide more accurate correlations between the bulk thermomechanical properties and rain erosion resistance of candidate LEP materials for wind turbine blade applications. Such correlations were provided from novel DMA experimental procedures in the context of the application, and the introduction of a combined DMA+RET test method to permit intermittent testing and the monitoring of material properties with respect to the amount of droplet impacts.

It is envisaged that this test method will reduce the amount of RET that is typically required to screen development LEP coatings, by offering an alternative testing regime which is shortened and relatively inexpensive. This investigation has demonstrated that sufficient monitoring of key thermomechanical properties may be obtained in less than one hour of rain erosion testing, in contrast to the standard route outlined within DNVGL-RP-0171 that may require between 20–200 h of RET and accompanying labour and asset costs. This reduction in testing time therefore offers greater flexibility to coating formulators and an increased product development rate.

Previous research has sought to infer property–erosion correlations by the measurement of relevant material properties prior to RET. This journal article aims to highlight greater clarity on key material properties due to the monitoring of such properties before and after exposure to droplet impacts. Furthermore, application of the DMA+RET test method has enabled differentiation and effective performance screening between candidate LEP coatings of equivalent polymer composition. The majority of property–erosion correlations proposed to date have compared polymers of broadly different chemical functionalities, in which measurable differences in polymer properties are subsequently more widely observed. However, many of these polymer classes, e.g., PBT, PE and PC, are not utilised as LEP products due to their relatively poor rain erosion resistance and are therefore not relevant to product developers. This investigation has therefore provided further evidence towards specific material properties that are proposed to be correlated with rain erosion resistance in terms of a causal relationship, i.e., cause and effect, in contrast to

material properties that may be indirectly correlated to rain erosion resistance, and which demonstrated negligible causation in this study, e.g., storage modulus or ETB.

In terms of the coating systems studied within this investigation, it is proposed that PA-U exhibits superior rain erosion resistance for two primary reasons, namely relatively high viscoelastic moduli (E′ and E″) and damping ability (tan δ) across a greater range of applied stresses that may be equivalent to raindrop impact, and improved short-term recovery. The viability of using polyaspartates as a rain erosion resistant material within LEP products, in addition to the synthesis and application of a novel LEP coating, has also been proven due to the minimal mass loss observed after 30 h of RET, which compares favourably with commercial LEP products that are based on polyurethane and polyurea functionality.

The relative importance of a LEP coating exhibiting increased E″ properties in tandem with a high tan δ value to maximise rain erosion resistance is highlighted and considered to be of greater importance than the E′, as observed for a given material composition (in this case PA-U when compared to PA-FB and PA-FS). The requirement of maximising E″ and tan δ will also inevitably increase the E′ of a material from a mathematical perspective. However, it is not suggested that an increase in E′, in isolation, is recommended for improving rain erosion resistance. Both factors (damping ability and short-term recovery) effectively reduce the probability of plastic deformation and coating microstructure damage from occurring after repeated raindrop impacts.

Furthermore, the presented results demonstrate that greater emphasis should be placed on creep recovery and non-linear strain testing for viscoelastic polymeric LEP materials. It has been demonstrated that solely performing oscillatory DMA experiments within the LVER of candidate materials may not sufficiently differentiate their respective rain erosion resistance. A holistic approach of performing several experimental procedures by DMA and combined DMA+RET is an effective screening approach for LEP formulations prior to performing RET in some format. This is due to the likelihood of LEP products being subjected to both linear and non-linear strains across various rainfall intensities in service.

**Supplementary Materials:** The following supporting information can be downloaded at: https://www.mdpi.com/article/10.3390/coatings13111849/s1, Figure S1: Tensile stress-strain curves of 3M Wind Protection Tapes (W8751 and W8607) in comparison to candidate polyaspartate binder formulations; Table S1: Tensile properties of 3M Wind Protection Tapes (W8751 and W8607) in comparison to candidate polyaspartate binder formulations; Figure S2: Representative DMA creep recovery curve where applied stress = dashed line; and resulting strain curve = solid line.

**Author Contributions:** Conceptualization, K.D. (DMA+RET) and C.S. (coatings); methodology, S.M.J.; investigation, S.M.J. and C.S.; writing—original draft preparation, S.M.J. and C.S.; writing—review and editing, K.D and N.R.; supervision, K.D. and N.R.; project administration, N.R. All authors have read and agreed to the published version of the manuscript.

**Funding:** This research was funded by Horizon 2020 (grant agreement ID: 953192) under the "Industrial Leadership–Leadership in enabling and industrial technologies–Advanced materials" programme as part of the Carbo4Power project.

**Data Availability Statement:** Data will be made available on request.

**Conflicts of Interest:** The authors declare no conflict of interest.

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
