# Peer review of "The Development of a Novel Thin Film Test Method to Evaluate the Rain Erosion Resistance of Polyaspartate-Based Leading Edge Protection Coatings"

_coatings, doi:10.3390/coatings13111849_

Round 1
Reviewer 1 Report
Comments and Suggestions for Authors
Comments to the Authors
I have gone through the manuscript entitles “The development of a novel thin film test method to evaluate 2the rain erosion resistance of polyaspartate-based leading edge 3protection coatings”. The authors have done good work, writing is good but need to reduce the introduction, and materials and methods parts. My suggestion is to make a separate supplementary file to reduce these parts. The authors need to explain the reason behind the increase in storage/loss modulus with respect to frequency.
Author Response
Reviewer 1,
Thank you for taking time to review our manuscript. We have incorporated the following changes to the manuscript based on your comments:
- The data and analysis relating to the tensile testing and initial screening of polyaspartate binders within subsection 2.1 – Tensile testing and material selection (previously Lines 147-170), has been included in a newly created Supplementary Materials section. A portion of the text and accompanying diagram used to describe the theory and method of the DMA creep recovery experimental procedure within subsection 2.2.2 – Combined DMA and RET (previously Lines 242-272), has also been included in the Supplementary Materials section.
- An explanation regarding the observed increase in storage and loss modulus with respect to frequency has been provided with accompanying references (Lines 316-320).
Regards,
Stephen Jones
Reviewer 2 Report
Comments and Suggestions for Authors
In this study, the authors present a novel thin-film test method designed to assess the
rain erosion resistance of polyaspartate-based leading edge protection coatings. The findings and their potential applications are compelling. However, several points need clear revision before this work can be considered for publication in 'Coatings':
1. Please include critical numerical values in the Abstract.
2. For Figure 2, differentiate the sections into Figure 2a and 2b.
3. Clarify the mechanisms (from molecular reactions to load testing) within the samples that pertain to the tensile stress-strain curves of the thin films. Furthermore, elucidate why these mechanisms lead to improvements.
4. The section "2.1. Tensile testing and material selection" appears twice, on Pages 4 and 6. Notably, section “2.2” is missing.
5. If applicable, cite a source for Figure 5.
6. Provide a citation for Equation (1).
7. The definition of 'recovery time' is necessary.
8. If feasible, compare the mentioned "recovery time of 10 minutes" with previously published works. Such a comparison will help readers understand if this duration is relatively short or extended.
9. For Figure 7, ensure clear scale bars are included.
10. Elaborate on the reason for the decrease in loss modulus with increasing oscillation strain.
11. Lastly, the recoverable compliance of the PA-U sample was observed to be the least among all the samples. What is the underlying cause? Kindly discuss the associated mechanism.
Comments on the Quality of English LanguageModerate editing of English language required
Author Response
Reviewer 2,
Thank you for taking time to review our manuscript. We have incorporated the following changes to the manuscript based on your comments:
- Critical numerical values have been included within the Abstract (Lines 23-27).
- Figure 2 (now Figure 1) has been separated into two sections: Figure 1a and 1b (Lines 176-178).
- The data and analysis relating to the tensile testing and initial screening of polyaspartate binders within subsection 2.1 – Tensile testing and material selection (previously Lines 147-170), has been included in a newly created Supplementary Materials section at the request of another reviewer. Further text and analysis has been added in context to the stress-strain curves of the polyaspartate-based coatings and their relevancy to performance as LEP products (Lines 26-34 in SM).
- This typing error has been corrected. Section 2.2 now includes the title Experimental procedures (Line 179).
- The photograph in Figure 5 (now Figure 4) is taken from an internal source (to Offshore Renewable Energy Catapult). This information has been included within the caption for Figure 4 (Lines 207-208).
- A portion of the text and accompanying diagram used to describe the theory and method of the DMA creep recovery experimental procedure within subsection 2.2.2 – Combined DMA and RET (previously Lines 242-272), has also been included in the Supplementary Materials section at the request of another reviewer. Equation 1 is now titled Equation S1 and a reference has been added (Line 57 in SM).
- 8. A definition of recovery time with context to the DMA experimental procedure has been added (Lines 244-251) and a referenced short-term recovery time has also been referenced and commented upon (Lines 98-106).
- Scale bars have been included for Figure 7 (now Figure 5) (Lines 269-270).
- An elaboration (with reference added) on the observed decrease in E’’ with respect to increased oscillation strain is provided (Lines 361-363).
- A discussion of the associated correlation between coating formulation and recoverable compliance/elasticity is included (Lines 395-405).
Regards,
Stephen Jones
Reviewer 3 Report
Comments and Suggestions for Authors
This work presented a useful method to evaluate the rain erosion resistance of polyaspartate-based leading edge protection coatings. However, I have some question listed as below:
1 It is suggested to add a schematic diagram including the sample preparation, equipment and the test results.
2 Whether the dimensions of 50 mm×10mm is a standard size. If not, please illustrate it.
3 Please confirm the specific meanings of E', E'', and tan δ in the text in order to fully understand the results described by the author.
4 Please confirm whether the correction of "PA-U" to "PA-FB" should be made in line 356 of the article.
5 From Figure 7, it can be seen that there are significant differences in surface defects among the three samples in the second stage of rain erosion. The coating starts exhibiting significant and accelerated mass loss, leading to surface roughness until failure occurs. Is this definition appropriate for the exposure of laminated composite plates? Will there be significant experimental errors in the test?
6 The quality of all data figures can be improved.
7 The title of Section 2.2 is typo and repeated with Section 2.1. Please check the language carefully.
8. The innovation and the advantage of the proposed metho can be enhanced.
Comments on the Quality of English LanguageThe language quality can be improved.
Author Response
Reviewer 3,
Thank you for taking time to review our manuscript. We have incorporated the following changes to the manuscript based on your comments:
- Please could you elaborate on this comment regarding the content of a schematic diagram in addition to what is currently illustrated within the manuscript i.e. Figure 1 – Cross-section of experimental setup; Figure 2 – Sample preparation; Figure 3 – Test specimens; Figure 4 – Test setup. Did you envisage a flowchart or process diagram to outline the sequence of steps taken to obtain the test results?
- The free film dimensions of 50 mm x 10 mm are considered as standard when performing DMA testing. Additional text has been added to the manuscript to reiterate this point (Lines 213 – 215).
- Definitions of E’, E’’ and tan δ have been included within the Introduction section and additional context has been provided with reference to the literature (Lines 121-130).
- This correction is not required to be made.
- The definitions for incubation period and end of incubation period were selected from DNVGL‑RP‑0573 recommended practice for the evaluation of erosion and delamination for leading edge protection systems of rotor blades. A reference to this document has been added to the manuscript (Line 262). This document specifies that the end of incubation period is defined as the time when the first mass loss or damage is visually detectable. The definition is therefore applicable to the evaluation of leading edge protection products e.g. coatings/tapes/erosion shields that are coated or adhesively bonded to a composite substrate as presented in this manuscript, but not solely applicable to the exposure of laminated composite plates. It is recommended that this test is repeated a minimum of three times per sample (which has been performed in this investigation). Error bars are also included on a revised Figure 6 to illustrate the standard deviation in the mass loss measurements.
- Please could you elaborate on this comment as to how we may improve the quality of all data figures.
- This typing error has been corrected. Section 2.2. now includes the title Experimental procedures (Line 179).
- Further text regarding the innovation and comparable advantages of the test method and wider investigation have been included at several points in the Abstract and Section 4 – Conclusions:
a) This test method is innovative as it has enabled the only documented occurrence of material property measurement before and after raindrop impact. It is believed that all prior scientific literature in this research area has solely performed material property measurements prior to rain erosion testing. This has provided further evidence regarding the key material properties that are required for rain erosion resistance in LEP products (Lines 21-23 and Lines 440-443).
b) This DMA+RET test method offers a significant reduction in RET time and associated costs in comparison to standard RET, which aids formulation development and innovation of LEP products (Lines 432-439).
c) Previous research has inferred material-property correlations based on test data when comparing polymers of significantly different functionalities that are not currently used as LEP products due to their poor rain erosion resistance. This investigation has therefore provided further evidence towards specific material properties that are proposed to be correlated with rain erosion resistance in terms of a causal relationship i.e. cause and effect, in contrast to material properties that may be indirectly correlated to rain erosion resistance and which demonstrated negligible causation in this study e.g. storage modulus or ETB (Lines 443-454).
d) The viability of using polyaspartates as a rain erosion resistant material within LEP products and synthesis/application of a novel LEP coating material has also been proven due to the minimal mass loss observed after 30 hours of RET, which compares favourably with commercial LEP products that are based on polyurethane and polyurea functionality (Lines 459-463).
Regards,
Stephen Jones
Reviewer 4 Report
Comments and Suggestions for Authors
The authors present a combined test method based on the iterative use of dynamic mechanical analysis (DMA) and rain erosion testing (RET) to understand which coating material properties strongly correlate with the better performance and durability of the coating polymer. Previous studies were largely aimed at defining material properties that provide better erosion resistance of that material. A combined testing method developed in the current study, however, allows to investigate the changes in material behavior during RET. In other words, the combined testing monitors several key material properties concerning raindrop impact, which provides valuable information about the property deterioration in the polymer under the ret test.
In principle, the idea of continuously tracking property deterioration under the ret is a good idea, however, such combined testing should include more robust testing parameters such as,
-thermal analysis because the blades are usually exposed to low T,
-ultrasonic measurements,
-acoustic impedance characterization,
-UV exposure.
I also ask authors to discuss whether their hybrid method can include
-rain erosion lifetime performance prediction.
Separate paragraphs devoted to the discussion of the
-wear mechanism,
-the analysis of surface topography changes,
-the microstructure of the formulations
will be also helpful.
Without these data and related discussions the manuscript mostly looks like a technical report: it describes the methodology of the hybrid method, and confirms already known and expected dependence: polymer formulation which exhibits the highest elasticity and potential for short-term recovery will have
superior raindrop erosion resistance.
My point is that a hybrid method is designed and tested on the already-known dependence. That dependence is fully expected and even can be derived using the ret method alone. I would like to see how the proposed method extends to take into account other factors (see above) while analyzing the reduction of plastic deformation and coating microstructure damage during raindrop impacts. In conclusion, in its current form, the manuscript reports already-known data regardless of the fact that the authors have developed a new testing mechanism. I ask the authors to extend their method by focusing on other material behaviors and show that their method can predict better formulations where the previous testing methods fail.
Author Response
Reviewer 4,
Thank you for taking time to review our manuscript. Please find attached responses to the specific points raised in your review:
1. “Combined testing should include more robust testing parameters such as thermal analysis because the blades are usually exposed to low T; ultrasonic measurements; acoustic impedance characterization; UV exposure.”
We are unable to provide additional data on combined testing within this manuscript. However, please find attached comments below regarding each aspect of combined testing:
- Thermal analysis: Thermal analysis of LEP materials during RET using the R&D A/S Rain Erosion Tester based at Offshore Renewable Energy Catapult indicated a 2 °C increase in temperature during operation under wet conditions. We therefore expect the impact of temperature during RET to be negligible. In terms of thermal analysis when performing DMA testing, we selected 20 °C as the nominal temperature as this represents a reasonable average monthly temperature of in-service blades when considering all Koppen climate systems. We also observed negligible changes in Tg before/after RET in LEP products tested to date.
- Ultrasonic measurements/acoustic impedance: Given that the speed of sound may be equated to the modulus of elasticity (c=SQRT(E/p)) and that we observed no significant change in elastic modulus before/after droplet exposure when measuring via frequency sweep/TTS, we have not performed ultrasonic/acoustic impedance measurements in this investigation.
- UV exposure: It is agreed that UV exposure (whether by means of outdoor weathering or accelerated weathering) will contribute to the rain erosion resistance and lifetime prediction of a LEP material. However, it is our understanding that there is no consensus within the scientific literature with regards to a recommended practice or exposure regime for candidate LEP products. We therefore believe that the effect of UV exposure upon the rain erosion resistance of LEP products is a highly significant research area and should be investigated in greater detail in isolation before it is combined with other characterisation techniques, in order to truly differentiate the significance of UV exposure upon the material properties of LEP products.
2. “…whether their hybrid method can include rain erosion lifetime performance prediction and separate paragraphs devoted to the discussion of the: wear mechanism; the analysis of surface topography changes; the microstructure of the formulations will be also helpful.”
We are unable to provide additional details on the analysis of surface topography changes or the formulation microstructure within this manuscript due to the required time constraints for resubmission. We do not believe this DMA+RET test method can include rain erosion lifetime performance prediction. At present, the most accurate tool for rain erosion lifetime prediction of LEP materials remains the analysis and interpretation of standard rain erosion test samples.
3. “…in its current form, the manuscript reports already-known data regardless of the fact that the authors have developed a new testing mechanism. I ask the authors to extend their method by focusing on other material behaviors and show that their method can predict better formulations where the previous testing methods fail.”
Further text regarding the innovation and comparable advantages of this test method and wider investigation in comparison to previous testing methods have been included at several points in the Abstract and Section 4 – Conclusions:
- This test method is innovative as it has enabled the only documented occurrence of material property measurement before and after raindrop impact. It is believed that all prior scientific literature in this research area has solely performed material property measurements prior to rain erosion testing. This has provided further evidence regarding the key material properties that are required for rain erosion resistance in LEP products (Lines 21-23 and Lines 440-443).
- This DMA+RET test method offers a significant reduction in RET time and associated costs in comparison to standard RET, which aids formulation development and innovation of LEP products (Lines 432-439).
- The viability of using polyaspartates as a rain erosion resistant material within LEP products and synthesis/application of a novel LEP coating material has also been proven due to the minimal mass loss observed after 30 hours of RET, which compares favourably with commercial LEP products that are based on polyurethane and polyurea functionality (Lines 459-463).
4. “…and confirms already known and expected dependence: polymer formulation which exhibits the highest elasticity and potential for short-term recovery will have superior raindrop erosion resistance. My point is that a hybrid method is designed and tested on the already-known dependence.”
In addition, further text has been included to address the key learnings on the topic of property-erosion correlations in Section 4 – Conclusions:
- One aim of the DMA+RET test method was to confirm the legitimacy of the existing property-erosion correlations that have been previously reported in the scientific literature. As detailed within Table 1 of the manuscript, there are up to 10 separately reported property-erosion correlations across various references and test methods. This previous research has inferred material-property correlations based on test data when comparing polymers of significantly different functionalities that are not currently used as LEP products due to their poor rain erosion resistance. This investigation has therefore provided further evidence towards specific material properties that are proposed to be correlated with rain erosion resistance in terms of a causal relationship i.e. cause and effect, in contrast to material properties that may be indirectly correlated to rain erosion resistance and which demonstrated negligible causation in this study e.g. storage modulus or ETB (Lines 443-454).
It is accepted that one of the prior references inferred a correlation between short-term recovery/elasticity and rain erosion resistance based on test data acquired using PMMA, PET, PC, PE and PP polymers. One conclusion of our investigation is that we believe this correlation to be valid and of significance for LEP materials. However, there are also numerous correlations previously listed within the scientific literature that state correlations and have been subsequently negated or disproven. We therefore believe it is valid scientific practice to perform further investigations that reaffirm and provide evidence to previously stated hypotheses when applied to novel or more relevant LEP materials, and that part of the novelty within this manuscript is the non-confirmation of other previously stated hypotheses, in addition to the novel aspects previously detailed in Point 3. We would also like to state that this hybrid method was not solely designed to test the dependence of short-term recovery upon rain erosion resistance. On the contrary, the inferred correlation of short-term recovery was one of the major conclusions as a consequence of applying this test method to the studied set of coating formulations. One advantage of the DMA+RET test method is the flexibility to screen multiple thermomechanical properties including equilibrium recoverable compliance but also E’, E’’, tan δ etc. The emphasis in this report was subsequently placed on the importance of short-term recovery because of the generated data.
Regards,
Stephen Jones
Round 2
Reviewer 2 Report
Comments and Suggestions for Authors
The manuscript can be accepted in this form.